# Estimation and Evaluation of Zenith Tropospheric Delay from Single and Multiple GNSS Observations

**Sai Xia** [1], **Shuanggen Jin** [1,2,3,*] and **Xuzhan Jin** [4]

1   School of Remote Sensing and Geomatics Engineering, Nanjing University of Information Science and Technology, Nanjing 210044, China; 20211211023@nuist.edu.cn
2   Shanghai Astronomical Observatory, Chinese Academy of Sciences, Shanghai 200030, China
3   School of Surveying and Land Information Engineering, Henan Polytechnic University, Jiaozuo 454003, China
4   College of Information Science and Engineering, Henan University of Technology, Zhengzhou 450001, China; 221170300105@stu.haut.edu.cn
*   Correspondence: sgjin@shao.ac.cn

**Abstract:** Multi-Global Navigation Satellite Systems (multi-GNSS) (including GPS, BDS, Galileo, and GLONASS) provide a significant opportunity for high-quality zenith tropospheric delay estimation and its applications in meteorology. However, the performance of zenith total delay (ZTD) retrieval from single- or multi-GNSS observations is not clear, particularly from the new, fully operating BDS-3. In this paper, zenith tropospheric delay is estimated using the single-, dual-, triple-, or four-GNSS Precise Point Positioning (PPP) technique from 55 Multi-GNSS Experiment (MGEX) stations over one year. The performance of GNSS ZTD estimation is evaluated using the International GNSS Service (IGS) standard tropospheric products, radiosonde, and the fifth-generation European Centre for Medium-Range Weather Forecasts (ECMWF) reanalysis (ERA5). The results show that the GPS-derived ZTD time series is more consistent and reliable than those derived from BDS-only, Galileo-only, and GLONASS-only solutions. The performance of the single-GNSS ZTD solution can be enhanced with better accuracy and stability by combining multi-GNSS observations. The accuracy of the ZTD from multi-GNSS observations is improved by 13.8%, 43.8%, 27.6%, and 22.9% with respect to IGS products for the single-system solution (GPS, BDS, Galileo, and GLONASS), respectively. The ZTD from multi-GNSS observations presents higher accuracy and a significant improvement with respect to radiosonde and ERA5 data when compared to the single-system solution.

**Keywords:** multi-GNSS; zenith tropospheric delays; GNSS meteorology; radiosonde; ERA5





## 1. Introduction

The tropospheric delay is an important factor affecting high-precision GNSS navigation positioning and timing [1]. Zenith tropospheric delay includes zenith hydrostatic delay (ZHD) that is related to the pressure, and zenith wet delay (ZWD) has high temporal and spatial variations in the atmosphere due to water vapor changes [2]. GNSS wet delays have shown to be a reliable technique for meteorological and climatological applications after two decades of experimentation and cross-checking with other approaches [3,4]. Compared to traditional techniques like radiosonde, GNSS can provide large-scale study and applications in meteorology with low cost, high sampling rate, and all-weather capability [5–7].

Nowadays, multi-GNSS systems, including GPS, BDS, Galileo, and GLONASS, have been well developed or updated [8–10], especially the rapidly developed BDS-3. The BDS-2 constellation was completed by the end of 2012, covering the Asia-Pacific region, and the BDS-3 constellation was completed by the end of 2020, providing global services. Nowadays, multi-GNSS systems have more constellations and more observations, with denser GNSS tracking stations worldwide providing a good opportunity to estimate ZTD.

The MGEX project was established by IGS in 2012, and can be used to track and analyze signals from multiple GNSS systems, including GPS, BDS, Galileo, and GLONASS

signals [11], and estimate coordinate, clock, and atmospheric products [12,13]. Experiment results showed that positioning precision and convergence rate can be improved by the integration of multiple GNSS systems with more visible satellites and observations when compared to the single-GNSS system [14,15]. For example, Li et al. [16] designed a four-system GNSS model, which resulted in a convergence time reduction of 70% and an accuracy improvement of 25% when compared to the single-GNSS system solution. With multi-GNSS combined processing, ZTD can be obtained with better accuracy within several millimeters [17]. Xu et al. [18] collected dual-frequency observations of GPS and BDS from some regions of China and compared BDS ZTDs with those of GPS. In [19], the authors showed that the accuracy of precipitable water vapor could be increased by a few millimeters when utilizing a GPS + BDS combined solution. The real-time water vapor of a four-system solution (GPS + BDS + Galileo + GLONASS) was also studied in [20] and demonstrated greater accuracy and stronger ZTD reliability when compared to the single-system solutions. Zhang et al. [21] used GPS and GLONASS observations to develop a tomography model and showed great potential in the study of water vapor profiles. Lu et al. [22] estimated real-time ZTD using real-time satellite orbit and clock products and the results were more reliable and accurate than that of the single-GNSS system. Lu et al. [23] developed multi-GNSS PPP ambiguity resolution for real-time ZTD retrieval which proved a great improvement in accuracy with respect to the Center for Orbit Determination in Europe, U.S. Naval Observatory products, and ECMWF data when compared to a GPS-only solution. Jiao et al. [24] analyzed the PPP performances of single-system and multi-GNSS combined PPP solutions and obtained a great improvement of the positioning accuracy and convergence by adding BDS-3 observations. The ZTD accuracy was improved by 20.5% when compared to the BDS-2. Alcay et al. [25] obtained the station coordinates by GPS, GPS + GLONASS, and GPS + GLONASS + Galileo + BDS combined solutions and the results showed that the ZTD differences between the three solutions were less than 20 mm. Farinaz et al. [26] used a network-based real-time kinematic approach to estimate ZTD by measurements of four constellations: GPS, Galileo, GLONASS, and BDS satellites. With an average of root mean squares error of roughly 12 mm, the ZTD showed great agreement with IGS products. Nzelibe et al. [27] proved a great improvement in accuracy in ZTD estimation for GNSS positioning when using the ERA5 atmospheric variables. Zhang et al. [28] proposed a method to obtain better ZHD corrections to solve the issues with the application of the conventional approach in areas with high variations in height, and their results showed that the method can achieve an improvement in accuracy of 50% over the conventional approach. ZTD estimation by early BDS was not good as other navigation systems due to regional BDS-2 or limited BDS-3 satellites. After the complete construction of BDS-3 since the end of 2020, the fully operating global BDS-3 constellation provides greater opportunity for GNSS meteorology.

In this paper, ZTD is estimated using the single- or multi-GNSS PPP technique over one year (2019) from 55 MGEX stations, particularly adding the fully operating BDS-3 constellation. Our main objective is to evaluate the performance of GNSS ZTD estimation from single- (GPS, BDS, Galileo, and GLONASS), dual-, triple-, or multi-GNSS observations. The multi-GNSS observations from 55 globally distributed stations are processed by Positioning And Navigation Data Analyst software version 1.0, which was developed originally by Wuhan University [29], and the performance of different ZTD solutions is evaluated with IGS troposphere products, radiosonde, and ERA5 hourly data [30].

## 2. Observation Model and Processing Strategy

A combination of dual-frequency carrier-phase and pseudo-range (LC, PC) is commonly used to eliminate the first-order ionospheric delay in PPP processing. The observation equation is given as follows [31],

$$L_{r,j}^s = \rho_{r\ g}^s - t^s + t_r + \lambda_j \left( b_{r,j} - b_j^s \right) + \lambda_j N_{r,j}^s - I_{r,j}^s + T_r^s + \varepsilon_{r,j}^s \tag{1}$$

$$P_{r,j}^s = \rho_{r\,g}^s - t^s + t_r + c\left(d_{r,j} - d_j^s\right) + I_{r,j}^s + T_r^s + e_{r,j}^s \tag{2}$$

where $s$, $r$, and $j$ denote GNSS satellite, receiver, and frequency, respectively; $\rho_g$ is the geometric distance; $t^s$ and $t_r$ refer to satellite and receiver clock offset; $\lambda_j$ is the wavelength; $b_{r,j}$ and $b_j^s$ are the uncalibrated phase delays; $T_r^s$ is the tropospheric delay; at different frequencies, the ionospheric delays $I_{r,j}^s$ can be expressed as $I_{r,j}^s = \kappa_j \cdot I_{r,1}^s$, $\kappa_j = \lambda_j^2 / \lambda_1^2$; and $N_{r,j}^s$ is the inter ambiguity; $e_{r,j}^s$ and $\varepsilon_{r,j}^s$ are the pseudo-range and carrier phase observations noises, which cannot be modeled; and $d_{r,j}$ and $d_j^s$ are the code biases. The offsets of antenna and phase wind-up can be rectified [32]. By interpolating the tidal constituents at each station according to Finite Element Solution 2004 and van Dam et al.'s models [33,34], oceanic and atmospheric tidal loading can be corrected. The GPS + BDS + Galileo + GLONASS observation model is given as follows in a multi-GNSS environment,

$$\begin{cases} L_{r,j}^G = \rho_{r\,g}^G - t^G + t_r + \lambda_{jG}\left(b_{rG,j} - b_j^G\right) + \lambda_{jG}N_{r,j}^G - \kappa_{jG}\cdot I_{r,1}^G + T_r^s + \varepsilon_{r,j}^G \\ L_{r,j}^C = \rho_{r\,g}^C - t^C + t_r + \lambda_{jC}\left(b_{rC,j} - b_j^C\right) + \lambda_{jC}N_{r,j}^C - \kappa_{jC}\cdot I_{r,1}^C + T_r^C + \varepsilon_{r,j}^C \\ L_{r,j}^E = \rho_{r\,g}^E - t^E + t_r + \lambda_{jE}\left(b_{rE,j} - b_j^E\right) + \lambda_{jE}N_{r,j}^E - \kappa_{jE}\cdot I_{r,1}^E + T_r^E + \varepsilon_{r,j}^E \\ L_{r,j}^{R_k} = \rho_{r\,g}^R - t^R + t_r + \lambda_{jR_k}\left(b_{rR_k,j} - b_j^R\right) + \lambda_{jR_k}N_{r,j}^R - \kappa_{jR_k}\cdot I_{r,1}^R + T_r^R + \varepsilon_{r,j}^R \end{cases} \tag{3}$$

$$\begin{cases} P_{r,j}^G = \rho_{r\,g}^G - t^G + t_r + c\left(d_{rG,j} - d_j^G\right) + \kappa_{jG}\cdot I_{r,1}^G + T_r^G + e_{r,j}^G \\ P_{r,j}^C = \rho_{r\,g}^C - t^C + t_r + c\left(d_{rC,j} - d_j^C\right) + \kappa_{jC}\cdot I_{r,1}^C + T_r^C + e_{r,j}^C \\ P_{r,j}^E = \rho_{r\,g}^E - t^E + t_r + c\left(d_{rE,j} - d_j^E\right) + \kappa_{jE}\cdot I_{r,1}^E + T_r^E + e_{r,j}^E \\ P_{r,j}^{R_k} = \rho_{r\,g}^R - t^R + t_r + c\left(d_{rR_k,j} - d_j^R\right) + \kappa_{jR_k}\cdot I_{r,1}^R + T_r^R + e_{r,j}^R \end{cases} \tag{4}$$

where G, C, E, and R stand for the GPS, BDS, Galileo, and GLONASS satellites, respectively; $R_k$ is the GLONASS satellite with frequency k; and the code biases $d_{rG}$, $d_{rC}$, $d_{rE}$, and $d_{rR_k}$ are different in the receiver. It is crucial to consider the intersystem biases (ISBs), which refer to differences between systems, and the interfrequency biases (IFB), which are different between the receiver code biases $d_{rR_k}$ for GLONASS satellites with different frequencies when processing multi-GNSS combined observations. The GPS code bias is set to zero and the ISBs are in relation to the GPS satellites' biases.

Firstly, the precise satellite orbits and clocks are determined using the observation data of 55 stations, which are shown in Figure 1. It is important to note that for each system and for each GLONASS frequency, zero mean conditions for the ISB/IFB parameters are involved in the determination of satellite orbits and clocks. Due to short-term fluctuations, station coordinates and fixed satellite orbits allow the calculation and updating of satellite clocks every epoch. In the observation equations, the correlative terms can be eliminated when satellite orbits, clock corrections, and station coordinates are fixed, and the observation model could be interpreted as:

$$\begin{cases} l_{r,j}^G = t_r + \lambda_{jG}\left(b_{rG,j} - b_j^G\right) + \lambda_{jG}N_{r,j}^G - \kappa_{jG}\cdot I_{r,1}^G + T_r^s + \varepsilon_{r,j}^G \\ l_{r,j}^C = t_r + \lambda_{jC}\left(b_{rC,j} - b_j^C\right) + \lambda_{jC}N_{r,j}^C - \kappa_{jC}\cdot I_{r,1}^C + T_r^C + \varepsilon_{r,j}^C \\ l_{r,j}^E = t_r + \lambda_{jE}\left(b_{rE,j} - b_j^E\right) + \lambda_{jE}N_{r,j}^E - \kappa_{jE}\cdot I_{r,1}^E + T_r^E + \varepsilon_{r,j}^E \\ l_{r,j}^{R_k} = t_r + \lambda_{jR_k}\left(b_{rR_k,j} - b_j^R\right) + \lambda_{jR_k}N_{r,j}^R - \kappa_{jR_k}\cdot I_{r,1}^R + T_r^R + \varepsilon_{r,j}^R \end{cases} \tag{5}$$

$$\begin{cases} p_{r,j}^G = t_r + c\cdot d_{rG} + \kappa_{jG}\cdot I_{r,1}^G + T_r^G + e_{r,j}^G \\ p_{r,j}^C = t_r + c\cdot d_{rC} + \kappa_{jC}\cdot I_{r,1}^C + T_r^C + e_{r,j}^C \\ p_{r,j}^E = t_r + c\cdot d_{rE} + \kappa_{jE}\cdot I_{r,1}^E + T_r^E + e_{r,j}^E \\ p_{r,j}^{R_k} = t_r + c\cdot d_{rR_k} + \kappa_{jR_k}\cdot I_{r,1}^R + T_r^R + e_{r,j}^R \end{cases} \tag{6}$$

where $l_{r,j}^s$ is "observed minus computed" phase and $p_{r,j}^s$ is the pseudo-range observable. The tropospheric delay $T_r^s$ can be calculated from the hydrostatic components $Zh_r$, wet components $Zw_r$, gradients, and mapping functions:

$$T_r^s = Mh_r^s \cdot Zh_r + Mw_r^s \cdot [Zw_r + \cot(e) \cdot (G_N \cdot \cos(a) + G_E \cdot \sin(a))] \tag{7}$$

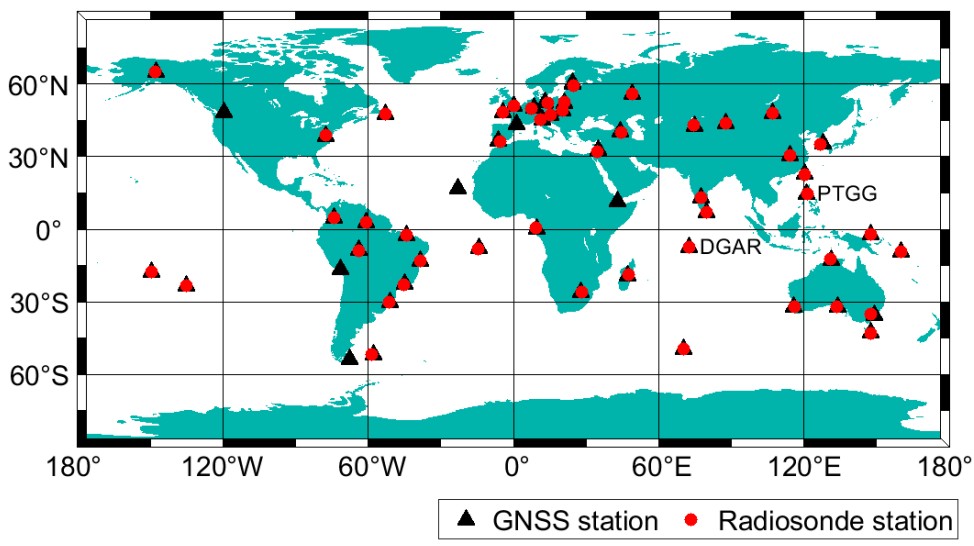

**Figure 1.** The distribution of GNSS stations from MGEX networks and radiosonde stations. The black triangle is the GNSS station, and the red circle is the radiosonde station.

With the Saastamoinen model and meteorological data, hydrostatic components $Zh_r$ can be calculated very accurately, while the wet components $Zw_r$ and gradients must be estimated as unknown parameters due to their variation in the atmosphere [35]; $Mh_r^s$ and $Mw_r^s$ are the hydrostatic and wet coefficients of the mapping function; $G_N$ and $G_E$ are the gradients in north and east directions; $e$ and $a$ are the elevation and azimuth angle.

By using an ionosphere-free linear combination, ionospheric delays are eliminated in time-consuming network solutions, greatly decreasing the number of calculated parameters. To enhance the PPP performance, we utilize the raw GNSS observation model with temporal-spatial ionospheric constraints [36]. The vector X for estimated parameters can be expressed as:

$$X = \left( Zw_r\, G_N\, G_E\, t_r\, d_{rE}\, d_{rC}\, d_{rR_k}\, I_{r,1}^s\, \overline{N}_r^s \right)^T \tag{8}$$

$$\overline{N}_r^s = N_r^s + b_r + b^s \tag{9}$$

In PPP processing, a sequential least squares filter is employed to estimate unknown parameters. The clock bias of receiver $t_r$ is calculated to be white noise. The ionospheric delays $I_{r,1}^s$ are estimated through dual-frequency raw pseudo-range and phase observations for each satellite at each epoch. The ISB and IFB parameters are calculated as constant. The phase ambiguity parameters $\overline{N}_r^s$ are calculated as constant for each continuous arc and can absorb the phase delays $b_r$ and $b^s$. The global mapping function (GMF) [37] is used to map tropospheric parameters. The tropospheric wet delay $Zw_r$ is modeled as a random walk process with the noise intensity of 5–10 mm/$\sqrt{hour}$ [17,22]. The stochastic model adopts the elevation-dependent weighting strategy $Q = 1/sin^2(ele)$, where *ele* is the elevation of the satellite. The elevation cut-off angle is set as 7°. Table 1 presents a summary of the multi-GNSS data processing strategy used for ZTD estimation.

**Table 1.** Multi-GNSS Data Processing Strategy for estimating ZTD.

| Item | Strategies |
|---|---|
| Estimator | Sequential least squares estimator |
| Observations | Raw carrier phase and pseudo-range observations; GPS + BDS + Galileo + GLONASS, about 126 satellites |
| Signal selection | GPS: L1/L2; GLONASS: L1/L2; Galileo: E1/E5a; BDS: B1/B2 |
| Sampling rate | 30s |
| Elevation cut-off | 7° |
| Weight for observations | Elevation-dependent weighting strategy |
| Receiver clock | Estimated, white noise |
| Satellite clock | Fixed |
| Satellite orbit | Fixed |
| ISB and IFB | Estimated as constant, GPS as reference |
| Phase wind-up effect | Corrected |
| Mapping function | GMF |
| Zenith Tropospheric delay | Initial modal + random walk model |
| Station displacement | Solid Earth tide, pole tide, ocean tide loading |
| Satellite antenna phase center | Corrected |
| Receiver antenna phase center | Corrected |
| Station coordinate | Fixed to coordinates of weekly solution |

## 3. Data Collection

### 3.1. GNSS Data

The IGS established the MGEX to track, collect, and analyze multi-system signals. Nowadays, the MGEX network has developed to over 400 stations with dense distribution, and most stations have the ability to receive signals from multiple GNSS systems. Finally, 55 GNSS stations, as shown in Figure 1, are used in this study, which can track four GNSS systems' signals. The observation data from each station are filtered, and each station is equipped with the capability to track signals from the GPS, BDS, Galileo, and GLONASS. The feasibility of parameters estimation for the single-system is ensured by the fact that each station has more than 80% data integrity and that more than six satellites can be observed for each navigation system. The estimation of tropospheric delay parameters by multi-GNSS is guaranteed to be accurate because of more than 20 observable satellites at each epoch. The redundancy of visible satellites can greatly improve the availability and reliability of estimated parameters.

### 3.2. Radiosonde Data

Radiosonde observations are obtained from the IGRA and GSL observations (https://ruc.noaa.gov/raobs/, accessed on 15 December 2022), which provide the vertical profiles of pressure, temperature, dew point, and wind parameters, among other meteorological variables [38]. Radiosonde can monitor and help understand the composition of lower atmosphere and track changes in water vapor fields for many decades. This observation technique is one of the most reliable techniques, which can study meteorological information of atmospheric vertical profiles [39]. However, it costs more than GNSS and other current techniques. Radiosonde data launched from dedicated stations using radiosonde balloons have low temporal resolutions, typically twice a day (0:00, 12:00 Coordinated Universal Time), and poor horizontal resolution, such as several hundreds of kilometers [40]. The radiosonde stations not far from the GNSS stations are marked by the red circle in Figure 1. The horizontal distance between the selected radiosonde stations and the GNSS stations is less than 25 km, and the altitude distance between them is less than 100 m to ensure that the deviation of ZTD between two different data sources is not large. Although there are many radiosonde stations, only a few that meet the criteria are selected. This paper unifies the ZTD at radiosonde station altitude to GNSS station using the GZTD2 model [41]. Table 2 displays the information of the nine radiosonde stations used, including the differences in height and distances between GNSS and radiosonde locations.

**Table 2.** Radiosonde stations used in this study.

| RS Station | GNSS Station | Elevation Difference (m) | Distance (km) |
|:----------:|:------------:|:------------------------:|:-------------:|
| 82281 | SALU | 16 | 2.1 |
| 82022 | BOAV | 67 | 1.6 |
| 91938 | THTG | 79 | 3.3 |
| 91948 | GAMB | 58 | 0.5 |
| 83971 | POAL | 47 | 10.0 |
| 57494 | WUH2 | −4 | 23.8 |
| 94610 | CUT0 | −7 | 9.7 |
| 94975 | HOB2 | −1 | 4.3 |
| 03882 | HERS | 26 | 3.8 |

The ZTD is obtained from radiosonde data by the integration method on different pressure layers. Due to the lesser ZWD above the top layer of the troposphere, the ZTD is equivalent to ZHD obtained by the Saastamoinen model [34]. The integration method is given as follows,

$$ZTD = ZTD_{level} + ZHD_{top} \tag{10}$$

$$ZTD_{level} = 1 \times 10^{-6} \int_{h_{Level}}^{h_T} N dh \tag{11}$$

$$N = k_1 \times \frac{P - e}{T} + k_2 \times \frac{e}{T} + k_3 \times \frac{e}{T^2} \tag{12}$$

$$e = \frac{q \times P}{0.622} \tag{13}$$

$$ZHD_{top} = \frac{2.2767 \times 10^{-3} \times P_T}{1 - 2.667 \times 10^{-3} \times \cos(2\varphi) - 2.8 \times 10^{-7} \times h_T} \tag{14}$$

where $ZTD_{level}$ represents ZTD below the top pressure level; $P_T$ and $h_T$ is the top pressure and height; $N$ is the refractivity; $P$ is the pressure; $e$ is the vapor pressure; $q$ is the specific humidity; $T$ is the temperature; $k_1$ = 77.604 K/hPa, $k_2$ = 64.79 K/hPa, $k_3$ =377,600.0 K/Pa; $\varphi$ is the latitude.

### 3.3. ERA5 Data

The ERA5 provides NWM products for atmospheric research. Currently, data are available from 1940 up to one week behind the real-time epoch [42]. Compared with previous ECMWF ERA-Interim reanalysis, the ERA5 has introduced significant improvement, such as hourly sampling, the increase in the spatial resolution from ~80 to ~25 km, and the temporal resolution from 6 to 1 h [43,44]. In this study, ERA5 layer products, downloaded from the ECMWF public dataset (https://cds.climate.copernicus.eu/cdsapp#!/dataset/, accessed on 2 February 2023), provide hourly temperature, geopotential, and specific humidity at 37 pressure levels with a horizontal resolution of 0.25°. Geopotential height was converted to geodetic height in order to achieve the same height unit [45–47]. The ERA5 ZTD can be obtained using the index of refractivity at each geopotential layer of the ERA5 data by the integration method [48].

### 4. Results and Analysis

*4.1. ZTD from Single-GNSS*

About 55 globally distributed stations from the MGEX network are processed in PPP mode with 30 s sampling interval to generate ZTD for the year 2019, where the simultaneous availability of GPS, BDS, Galileo, and GLONASS observations is required to examine the performance of the ZTD estimated from single-system and multi-GNSS combined solutions. The single-system (GPS, BDS, Galileo, GLONASS,), two-system (GPS + BDS, GPS + Galileo, GPS+GLONASS), three-system (GPS + BDS + Galileo, GPS + BDS + GLONASS, GPS + Galileo + GLONASS), and four-system (GPS + BDS + Galileo + GLONASS) solutions, are

adopted to derive ZTD. Weekly or biweekly fixes are made to a solution to correct the station coordinates.

In order to know the performance of ZTD estimation with adding BDS-3 observation data, the two stations, DGAR (7.3°S, 72.4°E) at Diego Garcia Island, UK and PTGG (14.5°N, 121.0°E) at Taguig City, Philippines, with the highest number of BDS annual observation satellites were selected. The time series of ZTD at the two stations are displayed in Figure 2. In general, the GPS-, BDS-, Galileo-, and GLONASS-estimated ZTD agree well, except for some outliers in the BDS-only solution. In Figure 2, significant fluctuations can be seen around 260 days at DGAR station and around 200 days at PTGG station. We investigated the BDS observation satellites during these periods and found that only three BDS-3 MEOs satellites were observed at DGAR station, while fewer than three BDS satellites were observed at PTGG station during DOY 199–202, which undoubtedly exacerbates the uncertainty of ZTD estimation in this region. This also indicates that the incomplete establishment of BDS-3 cannot cover all time periods in Southeast Asia. Compared to other navigation systems, GPS was developed earlier and has more full constellations. The most developed ZTD research is based on GPS. In order to analyze whether the ZTD estimated by the observation data of other constellation systems has a beneficial effect, this paper used the ZTD retrieved by GPS as a standard to explore the correlation with the ZTD from three other single systems. The BDS-derived ZTD data present larger noise and more outliers. ZTD scatter graphs between the GPS-only and the other single-system (BDS, Galileo, and GLONASS) solutions are shown in Figure 3 for stations DGAR and PTGG. At DGAR, the correlation coefficients of ZTD are 0.98, 0.99, and 0.99, and at PTGG, the correlation coefficients are 0.98, 0.99, and 0.99, implying that linear correlation between the GPS and other three systems is very high.

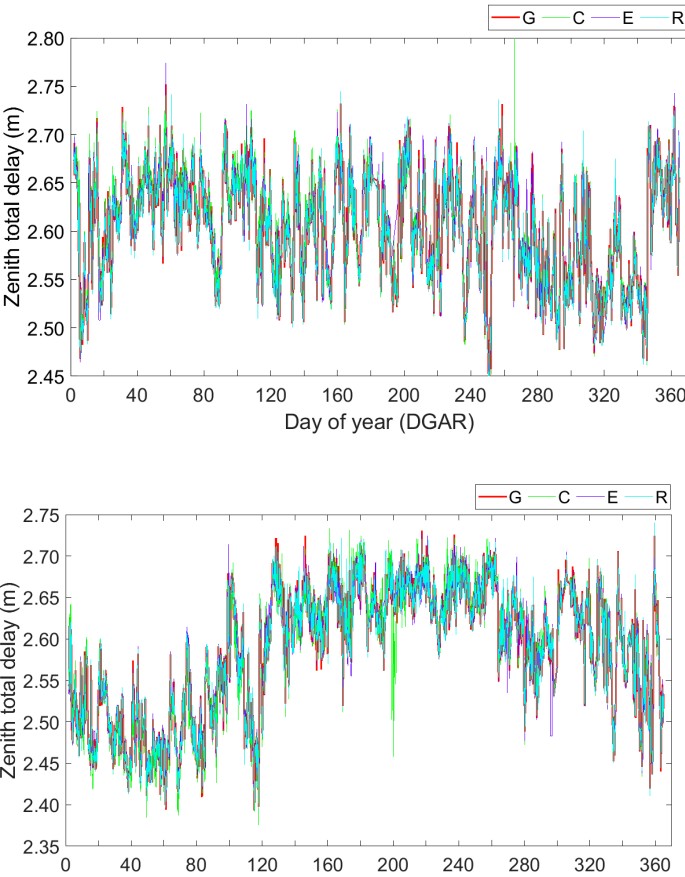

**Figure 2.** ZTD time series of GPS (G), BDS (C), Galileo (E), and GLONASS (R) for the year 2019 (**top panel**: DGAR; **bottom panel**: PTGG).

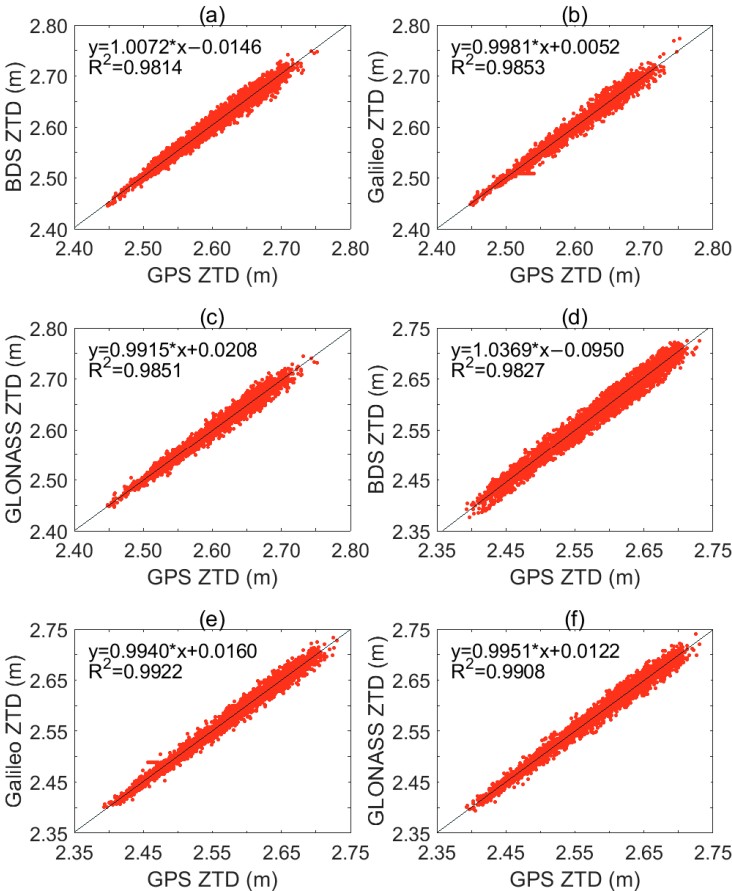

**Figure 3.** Linear correlation of GPS-derived ZTD to the other system-derived (BDS, Galileo, and GLONASS) ZTD at station DGAR (**a**−**c**) and PTGG (**d**−**f**).

The distribution of ZTD differences between the GPS-only and the other single-system solutions at DGAR and PTGG stations is shown in Figure 4. The ZTD differences in absolute values are mostly below 20 mm on average, and occasionally 20 to 30 mm. Between the GPS-only solution and the other single-system solutions, the root mean square (RMS) values of ZTD differences are 7.9, 5.9, and 6.3 mm for DGAR station, and 9.5, 6.4, and 6.8 mm for PTGG station, indicating good consistency within several millimeters. The mean biases of ZTD differences are −3.8, −0.2, and 1.3 mm for DGAR, and −0.4, −0.4, and 0.3 mm for PTGG.

The biases and RMSs of ZTD differences are shown in Figure 5 between the GPS-only and the other single-system solutions for 55 stations, excluding the value between GPS and BDS at BSHM station. Horizontal coordinates are sorted from the low to high latitude. Between GPS and GLONASS, the RMS value is the smallest, approximately 2.5–7.9 mm. It is greater between GPS and Galileo, ranging from 2.7 to 10.9 mm, and the greatest value between GPS and BDS, approximately 4.3–12.2 mm, respectively. Between GPS and Galileo, the RMS values are relatively large at stations BREW and KERG. By examining the number of observable satellites for these two stations, we found that observable satellites for Galileo have averages of 2.6 and 3.0, respectively. However, the average number of observable satellites for GPS is 8 at both stations. The RMS of ZTD difference between GPS-only and GLONASS-only solutions is roughly equivalent with that between GPS-only and Galileo-only solutions. The possible reason is that the number of observable satellites is almost the same, with an average number not exceeding 1.5. BDS is significantly different from the other three systems; BDS-3 has not been fully established, and the limited number of observed values inevitably leads to significant errors in positioning and estimating parameters. This confirms that the number of observable satellites in one system greatly

affects the accuracy of parameter estimation. The biases are mostly from −2 to 2 mm, and the parts with large deviation values are almost all between GPS and BDS. In general, the RMSs show a decreasing trend from the low latitude to high latitude.

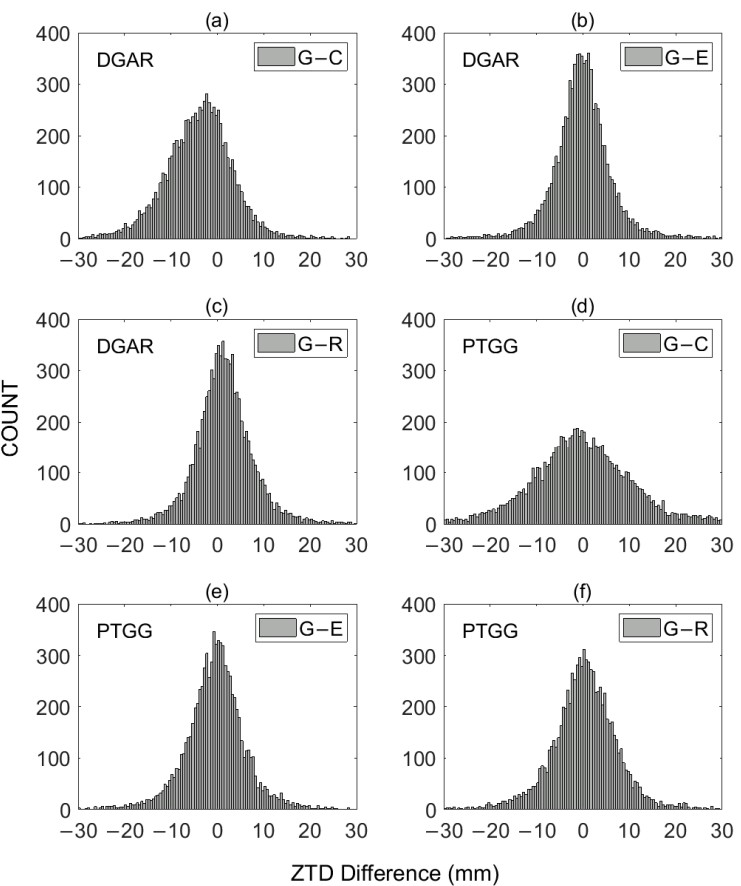

**Figure 4.** Distribution of ZTD differences between the GPS-only and the other single-system (BDS, Galileo, and GLONASS) solutions at stations DGAR (**a**−**c**) and PTGG (**d**−**f**).

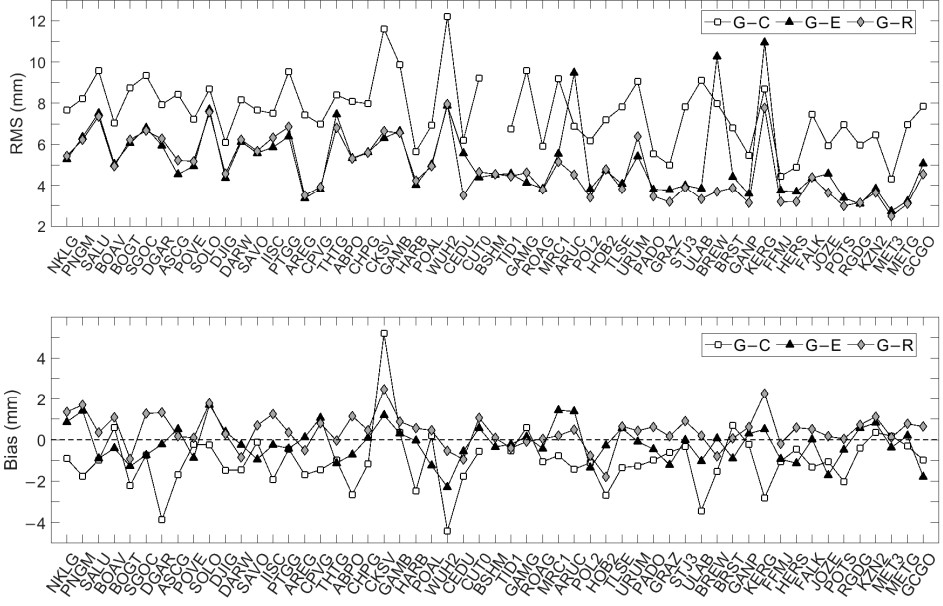

**Figure 5.** Between the GPS-only and the other single-system (BDS, Galileo, and GLONASS) solutions, the top panel shows the RMSs of ZTD differences and the bottom panel shows biases of ZTD differences.

### 4.2. ZTD from Multi-GNSS

ZTD differences between four-system combined and other solutions at stations DGAR and PTGG in the year 2019 are displayed in Figure 6. Taking the four-system solution as a reference, it is simpler to see the improvement of ZTD's stability and discreteness from single-system to two-system, then to three-system solutions. In the single-system solution, it is evident that the GPS difference is the lowest and most stable relatively; the Galileo difference is larger than GPS, although a tiny quantity of the data are missing. The ZTD difference for GLONASS is larger than that for Galileo, with consistent data but larger fluctuations. The BDS difference has the greatest fluctuations and is the largest. ZTD differences at station DGAR show different levels of fluctuation. For example, overall ZTD differences are larger than 0 mm during the period of DOY 0−130 and smaller than 0 mm during the period of DOY 130–270. At DGAR station, ZTD estimation of multi-GNSS solutions cannot guarantee high accuracy at all times, but adding more satellite observations can indeed greatly improve accuracy and stability compared to the single-system solution. We also found significant fluctuations at DGAR station, DOY 282 for GE, GCE, and CER, possibly due to quality problems with Galileo's data in some cases.

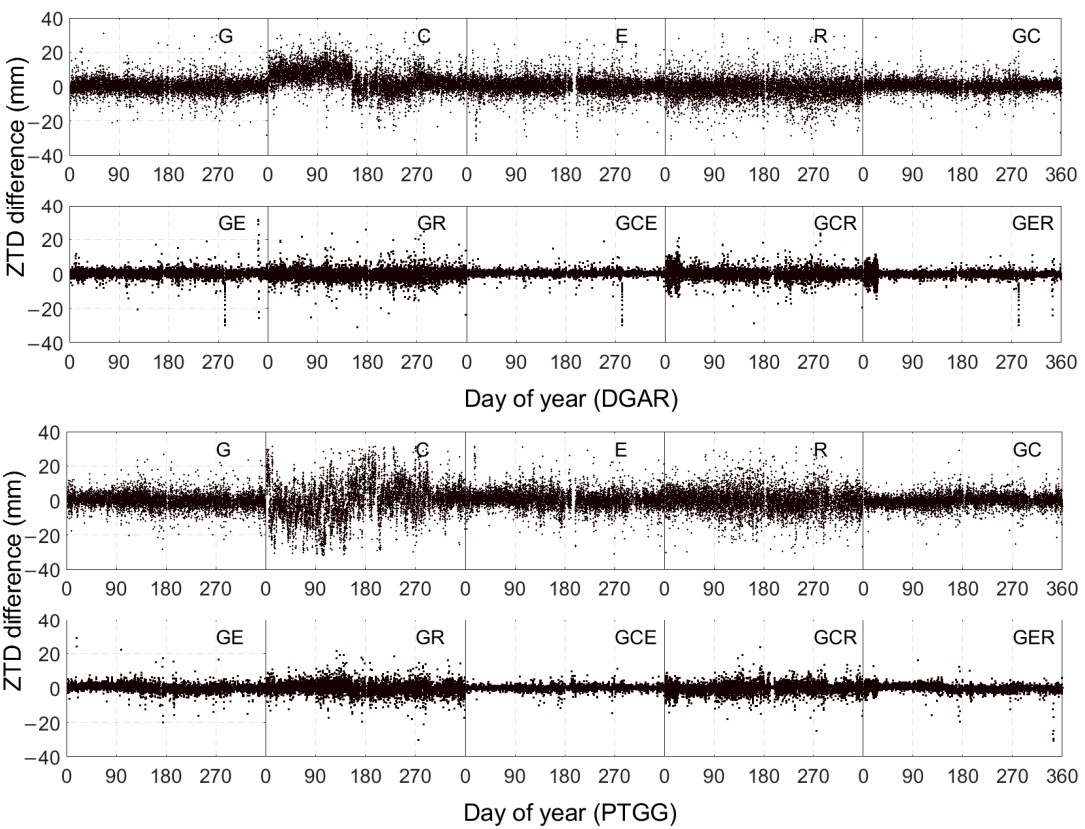

**Figure 6.** ZTD differences between four-system combined and single-system (G, C, E, and R) or multi-system combined (GC, GE, GR, GCE, GCR, and GER) solutions (**top panel**: DGAR; **bottom panel**: PTGG).

It can be clearly seen that the two-system solution has greatly improved when compared to the single-system solution. The difference for GPS and Galileo combined solutions is the smallest, while that for GPS and GLONASS is larger and more discrete. Overall, the three-system combined solutions present better results than two-system solutions, with the GCE combined solution being the best relatively, followed by the GER combined solution. Although the GCR combined solution is somewhat inferior, it is much better than GC and GR combined solutions. Probable reasons are that some observations are not yet available because of insufficient visible satellites and poor data quality in some cases. The RMS values for single-system (G, C, E, and R) are 3.0, 7.6, 4.0, and 4.2 mm; for two-system

(GC, GE, and GR) are 2.6, 1.5, and 2.3 mm; and for three-system (GCE, GCR, and GER) are 1.0, 2.2, and 1.3 mm, respectively. When there is data loss in a single system during certain time periods, a combination of multiple systems can compensate for this flaw, which means that in GNSS Meteorology research, receivers that can receive signals from multiple systems have significant advantages. High robustness and availability can be guaranteed for ZTD estimated from multi-GNSS fusion since the four-system ZTDs are more consistent and stable.

### 4.3. Evaluation by IGS Final Products

To evaluate the precision of ZTD obtained from different solutions, we use three different types of reference data: IGS final troposphere products, radiosonde data, and ERA5 data. The sample interval of the three different independent validation data types differs, and therefore temporal and spatial matching are required and the amount of data is sufficient to ensure that interpolation on results of the evaluation is not needed.

The time series of ZTD obtained from single-system and multi-system solutions, so as to compare with IGS final troposphere products at stations DGAR and PTGG, are shown in Figure 7. The ZTD values of the BDS-only solution present larger values, relatively, and those of multi-system solutions agree very well with those from IGS tropospheric products (within several millimeters).

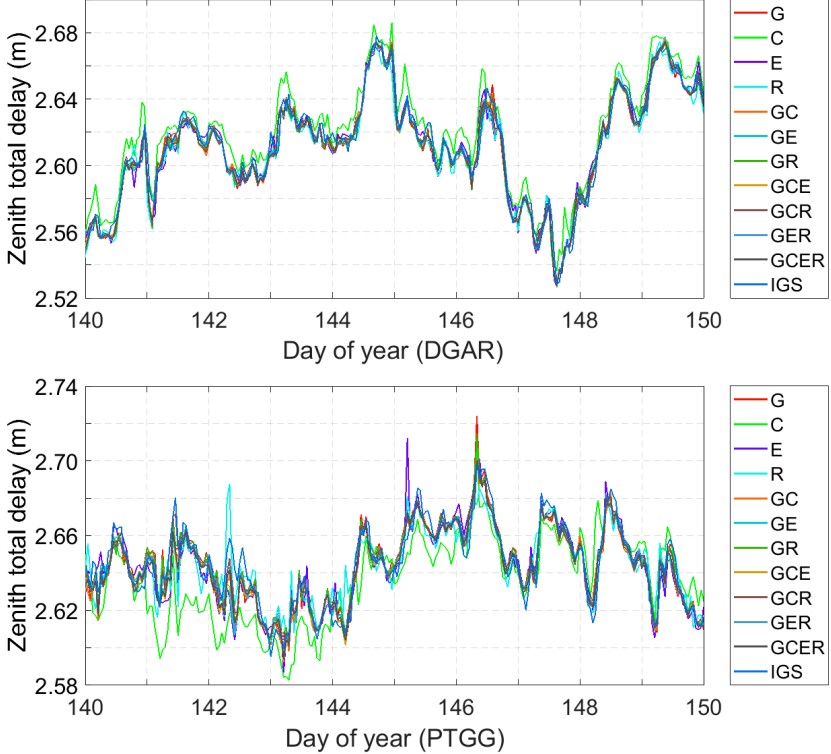

**Figure 7.** ZTD derived from single-system and multi-system solutions and IGS final troposphere products during DOY 140–150, 2019 (**top panel**: DGAR; **bottom panel**: PTGG).

The ZTD differences of single- and multi-system solutions with respect to IGS products at stations DGAR and PTGG, are shown in Figure 8. At DGAR, ZTD differences of BDS-only are mostly above 0 mm, and those of other solutions appear stable with few fluctuations. At PTGG, the ZTD differences of the BDS-only solution are mostly below 0 mm, and those of the GLONASS-only solution present a significant fluctuation. It is worth noting that there are few large fluctuations in the multi-system combined solutions. The differences of single-system solutions mainly range from −20 to 20 mm and those of multi-system combined solutions are smaller than 10 mm.



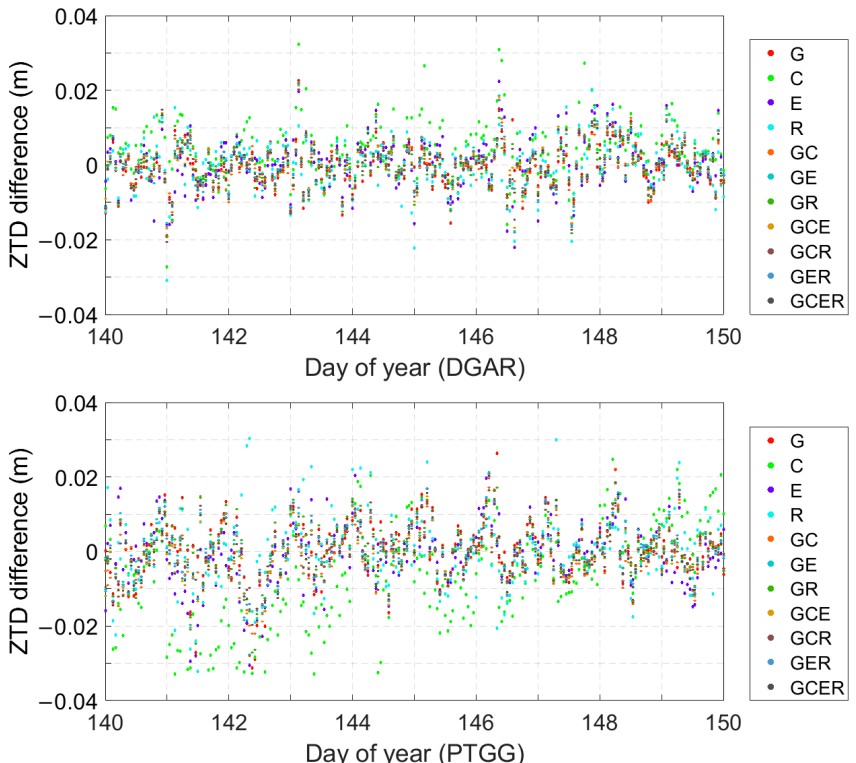

**Figure 8.** The ZTD differences of single- and multi-system solutions with respect to IGS products during DOY 140–150, 2019 (**top panel**: DGAR; **bottom panel**: PTGG).

Figure 9 displays the RMS and mean absolute bias values for ZTD differences of single-system and multi-system solutions with respect to IGS products at three stations, including DGAR, PTGG, and AREG. Other stations equipped with four-system GNSS receivers and IGS products are also considered. For the four-system solution, the RMS values are about 3.9–6.3 mm and for the single-GNSS solutions (G, C, E, R), the values are about 4.7–9.1 mm, 5.6–12.8 mm, and 4.9–12.1 mm, respectively. The absolute bias values are also reduced by about 1–3 mm. The difference between two-system and three-system solutions is small, which can also be seen in Figure 6. In other words, the degree of accuracy improvement is not as good as that between single-system and two-system solutions. The possible reason is that the observations reach a certain number, and the accuracy improvement of the unknown parameters' estimation is basically not significant. In order to obtain a higher accuracy estimation, increasing the number of observations will also lead to longer data processing time.

Taking the IGS products as a reference, the RMS values for the two-, three-, and four-system solutions are reduced by 2.8%, 5.3%, and 13.8% respectively, compared to the GPS-only solution when including GPS observations in the PPP solution. Similarly, RMSs of the two-, three-, and four-system solutions fall by 35.1%, 37.2%, and 42.8% respectively, compared to the BDS-only solution when including BDS observations in the PPP solution. RMSs of the two-, three-, and four-system solutions are also decreased by 17.5%, 20.1%, and 27.6% respectively, compared to the Galileo-only solution when including Galileo observations in the PPP solution. The RMS values fall by 14.4%, 15.5%, and 22.9% compared to the GLONASS-only solution when including GLONASS observations in the PPP solution. The ZTD values produced for the GPS-only solution are the ones that are closest to IGS products from the standpoint of single-system solutions. From the perspective of the two-system solutions, the ZTD values obtained for the GPS/GLONASS combined solution are the most accurate. The ZTD values obtained for the GPS, Galileo, and GLONASS combined solution are the most precise in terms of three-system solutions. These results confirm that there is a significant decrease in RMS and absolute bias values in multi-system combined

solutions. Comparing the combined solutions to single-system solutions, ZTD estimates can be more accurate and reliable, and certain outliers appearing in single-GNSS solutions can be eliminated. These prove the significant potential of multi-system combination mesoscale atmospheric research and Numerical Weather Prediction models.

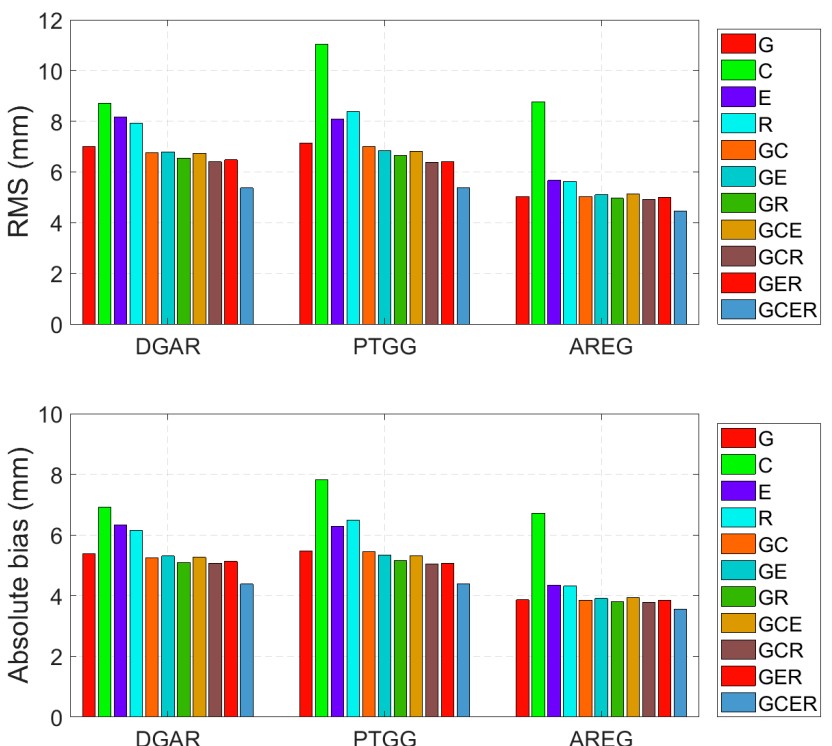

**Figure 9.** RMS and mean absolute bias for the ZTD differences of single- and multi-system solutions with respect to IGS final products (**top panel**: RMS; **bottom panel**: absolute bias).

### 4.4. Validation by Radiosonde and ERA5

Radiosondes are used to validate the ZTD results obtained from the multi-GNSS solutions. The ZTD is set to estimate once every 1 h while radiosonde-retrieved ZTD is 12 h. Consequently, we consider the ZTD from the same epochs when comparing.

Figure 10 illustrates a comparison of the ZTD results at stations POAL and HOB2 from the four-system solution solutions and nearby radiosonde solutions during DOY 80−180, 2019. The four-system combined ZTD is in considerable agreement with the ZTD from the radiosondes, with a disparity of a few millimeters, which implies the ZTD value can effectively be estimated by multiple systems at the bottom of the atmosphere. At nine GNSS stations, where radiosondes at a distance of 25 km or less can be found, RMSs of ZTD differences for single- and multi-system solutions in relation to the radiosonde solutions are shown in Figure 11. For some stations, such as POAL and CUTO, the RMS was reduced much from single-system to two-system solutions, but the improvement in accuracy is not very significant from two-system to four-system solutions, with an increase of 0.3–1.8 mm. Probable reasons are the good geographical location of some stations themselves, and in all ZTD values over a year, the abnormal ZTD values of different seasons will be diluted by other time periods. Therefore, increased multi-system observation can effectively solve this problem.

We can see that there are very few inconsistencies between the four-system solution and radiosondes. The RMS value decreases noticeably as the number of multi-system observation data increases. The average RMS of the ZTD differences for the GPS-only solution is 8.7 mm; for the two-system solutions, 8.3 mm; for the three-system solution, 7.8 mm; and for the four-system solution, 7.6 mm. Compared to the BDS-only solution, an improvement of 13–20% in accuracy can be achieved with the two-system, three-system,

and four-system solutions. The average RMSs of ZTD differences from multi-GNSS solutions are summarized in Table 3. This demonstrates the potential benefits of multi-system estimated ZTD in climatic research.

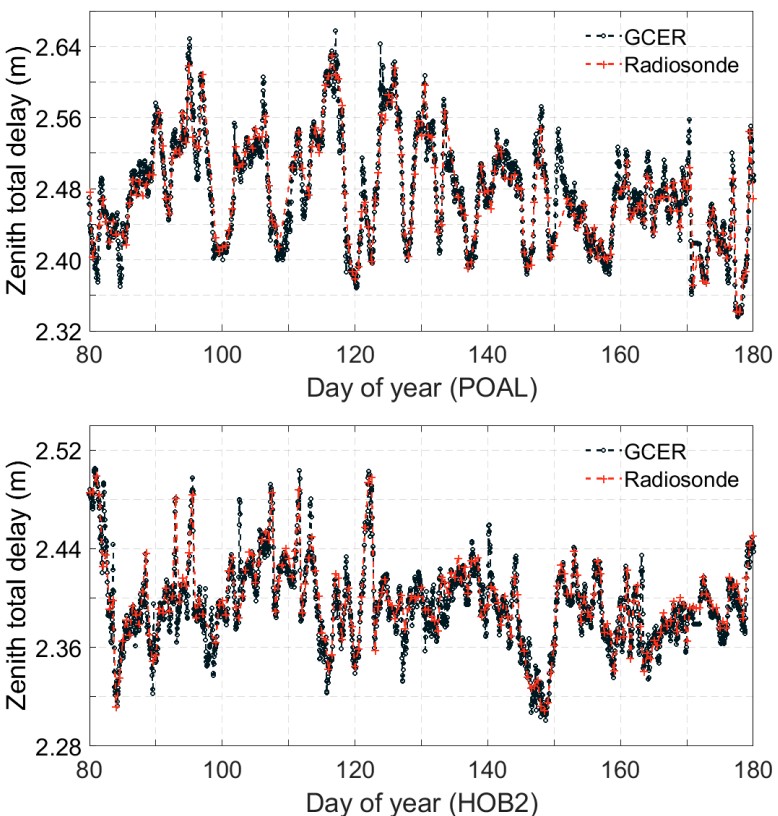

**Figure 10.** ZTD derived from the four-system solution and radiosonde solution during DOY 80−180, 2019 (**top panel**: POAL; **bottom panel**: HOB2).

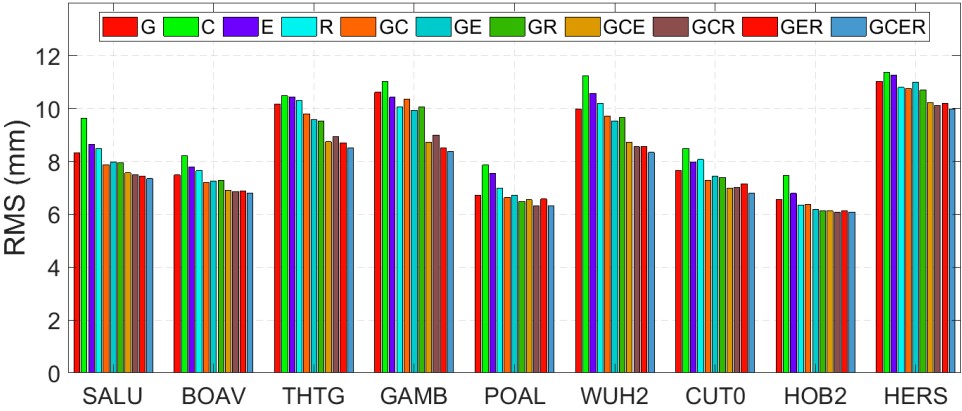

**Figure 11.** RMSs of ZTD differences for single- and multi-system solutions with respect to the radiosonde solutions.

In this analysis, the ZTDs generated by the ERA5 data are also used to validate those obtained from multi-GNSS processing. The time series of ZTD from multi-system solutions and ERA5 date at stations RGDG and STJ3 during DOY 120–160, 2019 are shown in Figure 12. Even if in some time periods, such as DOY 154–157 at station STJ3 and DOY 144–145 at station DGRG, where frequent changes in tropospheric water vapor cause significant fluctuations in ZTD time series, both multi-GNSS and ECMWF can correctly track these changes. RMSs of the ZTD differences with respect to the ERA5 data for single-system (G), two-system (GR), three-system (GER), and four-system (GCER) solutions are

calculated because the ZTDs of these solutions are more accurate and reliable in single-system, two-system, and three-system solutions; this is detailed in Section 4.3. Figure 13 shows that RMSs of ZTD differences are clearly decreasing from single-system to multi-system solutions and the average RMS values of stations are 10.6, 9.2, 8.7, and 7.8 mm, respectively. In general, high-latitude stations show better agreement in ZTD estimates than low-latitude ones in terms of geographical patterns. The average RMS value of stations for four-system above 30 latitudes is 6.8 mm while the average RMS in low-latitude (0–30 °) is 9.0 mm. From the distribution of stations on land and sea, no significant changes are found, or the feature is not easy to detect (possibly due to the fact that there are not enough stations with similar latitudes on the same continent). Applying the multi-system combined PPP processing strategy, even in low-latitude areas with frequent changes in tropospheric water vapor, the fluctuations in tropospheric delay can be well captured and corrected by the added multi-system observations.

**Table 3.** Average RMS and absolute bias of ZTD differences of single-system and multi-system solutions with respect to the radiosonde solutions at 9 stations.

| Solution | RMS (mm) | Bias (mm) |
|---|---|---|
| GPS | 8.7 | 5.1 |
| BDS | 9.5 | 5.6 |
| Galileo | 9.0 | 5.3 |
| GLONASS | 8.8 | 5.1 |
| GPS/BDS | 8.4 | 4.9 |
| GPS/Galileo | 8.3 | 4.8 |
| GPS/GLONASS | 8.3 | 4.9 |
| GPS/BDS/Galileo | 7.8 | 4.6 |
| GPS/BDS/GLONASS | 7.8 | 4.6 |
| GPS/Galileo/GLONASS | 7.8 | 4.5 |
| GPS/BDS/Galileo/GLONASS | 7.6 | 4.5 |

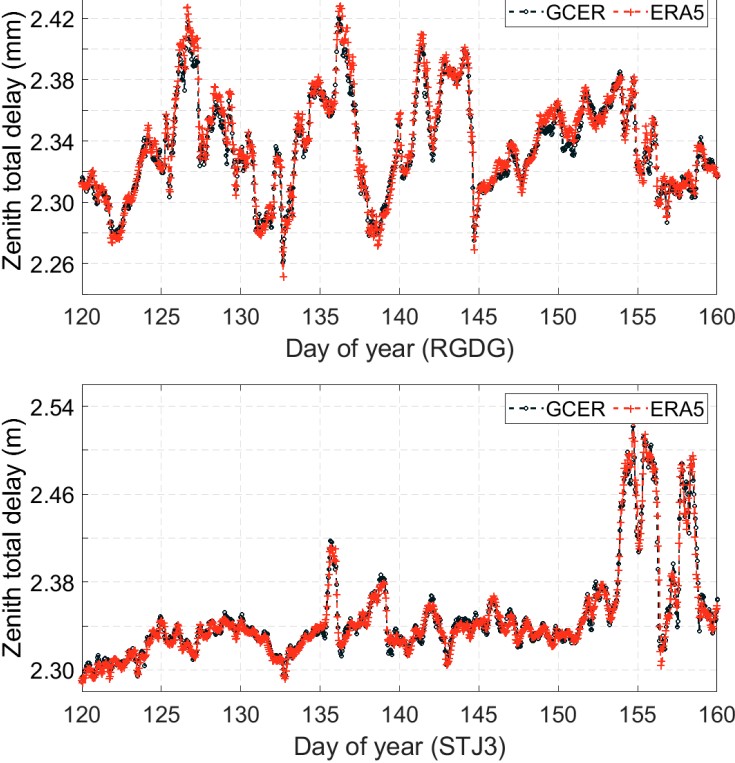

**Figure 12.** ZTD time series from the four-system solution and ERA5 data for a period of 40 days (**top panel**: RGDG; **bottom panel**: STJ3).

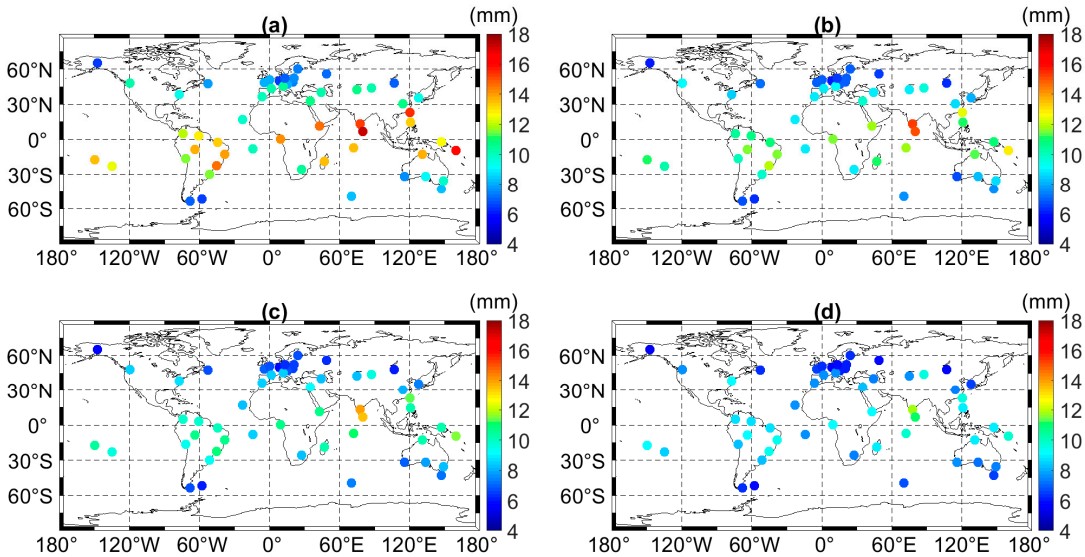

**Figure 13.** Geographical distribution of RMS values of ZTD differences for the multi-system solutions with respect to the ERA5 data at GNSS stations. (**a**) G. (**b**) GR. (**c**) GER. (**d**) GCER.

## 5. Discussion

The GNSS application in meteorology is becoming more and more popular with its rapid development. The increase in observation systems and observable satellites provides enormous data assistance for GNSS meteorological study and applications. Previous studies were done prior to or during the early stages of the BDS-3 satellite network with fewer BDS-3 satellites. Unlike previous studies, our study aims to use more data sources to validate ZTD values under different multi-GNSS combined strategies, especially the full operation BDS-3. For this purpose, we used multi-GNSS clock and orbit products to estimate the ZTD values and evaluated the performance of ZTD for multi-system solutions, as shown in Figures 2 and 3. The correlations between GPS ZTD and the ZTD of the other three systems are very high, but BDS ZTD shows significant fluctuations at some times. In Section 4.2, we compared the ZTD of the four-system solution with single-system, two-system, and three-system solutions. A great improvement of ZTD in accuracy and stability can be obtained by multi-GNSS solutions in Figure 6. In Sections 4.3 and 4.4, we evaluated the ZTD for different solutions when compared with IGS products, radiosonde, and ERA5 data. The results proved that an improvement of accuracy can be obtained by two or more systems when compared to the single-system solution.

With the establishment of more MGEX stations and the launch of more satellites, especially BDS-3, the number and quality of satellite observations have been improved. Therefore, higher accuracy in retrieving ZTD can be obtained from multi-GNSS observations, which will provide a good opportunity for GNSS atmospheric research and meteorological applications.

## 6. Conclusions

In this study, zenith tropospheric delay is estimated using the single-, dual-, triple-, or four-GNSS PPP techniques. The multi-GNSS observations of 55 globally distributed MGEX stations were used in single- and multiple-system solutions during the year 2019. The ZTD results are evaluated and compared with IGS products, radiosonde, and ERA5 data. The results show that, in general, the ZTD results of BDS, Galileo, and GLONASS are close to those of GPS. The RMS of the ZTD differences is about 4.3–12.2 mm between GPS-only and BDS-only solutions, about 2.7–10.9 mm between GPS-only and Galileo-only solutions, and about 2.5–7.9 mm between GPS-only and GLONASS-only solutions. The ZTD difference is not significant. Additionally, in the combined solutions, some outliers that appeared in single-GNSS solutions could be eliminated. With the increasing number

of satellites, ZTDs of the four-system solutions are more stable and consistent than those of the single-GNSS solutions.

The ZTD results from multi-GNSS combined solutions exhibit great consistency with IGS products. An improvement of 13.8%, 42.8%, 27.6%, and 22.9% in accuracy was achieved when compared to the GPS-only, BDS-only, Galileo-only, and GLONASS-only solutions by utilizing the multi-GNSS processing. The RMSs of the four-system solution are 5.5 mm with respect to IGS final products. The ZTD results from the four-system solution show a little deviation of a few millimeters with radiosonde data. This deviation in accuracy can be compensated for through the multi-GNSS solutions. The RMS of the GPS-only solution is 10.6 mm, whereas that of the four-system is 7.8 mm with respect to ERA5 data. Therefore, multi-GNSS-based tropospheric products with enhanced precision and dependability will greatly contribute to weather and climate change.

By validating multiple GNSS ZTD through multiple data sources, the accuracy of ZTD has been greatly improved from single-system to multi-system solutions. However, for some stations with good observations, the accuracy of ZTD is only slightly improved from two-system to four-system by adding more observations. The increase in observation redundancy, especially the establishment of BDS-3, will inevitably bring great convenience to improving the accuracy of positioning and atmospheric parameter estimation, which will also lead to problems such as longer data processing time and more costs. Therefore, it should improve the accuracy and efficiency of ZTD estimation from multi-GNSS observations together in the future.

**Author Contributions:** Conceptualization, S.J.; methodology, S.X. and S.J.; software, S.X.; validation, S.X. and S.J.; formal analysis, S.X.; investigation, S.X. and S.J.; resources, S.X.; data curation, S.X.; writing—original draft preparation, S.X., S.J. and X.J.; writing—review and editing, S.X., S.J. and X.J.; visualization, S.X. and S.J.; supervision, S.J.; funding acquisition, S.J. All authors have read and agreed to the published version of the manuscript.

**Funding:** This work was supported by the Open Fund Project of the Tianjing Key Laboratory for Rail Tranzit Navigation Positioning and Spatial-Temporal Big Data Technology (Grant No.: TKL2023A01).

**Data Availability Statement:** The GNSS observations and IGS final products can be obtained from https://cddis.nasa.gov/archive/gnss/ (accessed on 12 November 2022); the radiosonde data can be obtained from NOAA/GSL–RAOB (accessed on 15 December 2022); the ERA5 data can be obtained from https://cds.climate.copernicus.eu/cdsapp#!/dataset/ (accessed on 2 February 2023); the ZTD products can be obtained from the authors.

**Acknowledgments:** The authors would like to thank the IGS, NOASS, and ECMWF for providing the GNSS data and products.

**Conflicts of Interest:** The authors declare no conflict of interest.

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
