# Peer review of "Estimation and Evaluation of Zenith Tropospheric Delay from Single and Multiple GNSS Observations"

_remotesensing, doi:10.3390/rs15235457_

Round 1
Reviewer 1 Report
Comments and Suggestions for Authors
This manuscript comprehensively evaluates single, double, three, or four GNSS precision point positioning techniques. Moreover, the ZTD inverted from IGS, radiosonde, and ERA5 is used to evaluate the ZTD of the above-combined system technologies. The research results are helpful to understand better the application of different combinations of ZTD in meteorology. However, I pointed out several issues that must be addressed to improve the manuscript.

The quality of English language is good.
Reviewer 2 Report
Comments and Suggestions for Authors
1. In PPP processing, a sequential least squares filter is employed to estimate unknown parameters, that is, the tropospheric wet delay and horizontal gradient are modeled as a random walk process. In this case, RMS value ZTD will significantly decrease as the number of multi-GNSS observation data increases. This can be estimated "theoretically" and get the same % error reduction.
2. Adding more GMSS results in better sky coverage, especially for the low elevation and azimuth angles which leads to a better geometry for the future assimilation and tomography studies. Maybe the authors have another goal of their research?
3. The article does not say anything about the technology of radiosonde solutions and their evaluation. How did the authors practically obtain integral ZTD values and which models of the upper atmosphere were used for different latitudes. If we take into account the real estimates of radiosonde observations (near 10 mm), then the results of Table 3 do not confirm anything.
4. The authors performed a significant amount of experimental research. All their results were obtained by simply comparing one result with another. At the same time, there is no analysis of a priori errors of these results, which makes it difficult to understand the final conclusions.
5. Using some tropospheric product as a reference, the authors it has been shown that the GNSS PPP solutions have enhanced the precision of the ZTD estimates, in comparison with the single-system solution separately. However, accuracy changes when we choose some other tropospheric product as a reference (IGS standard troposphere products, radiosonde, Numerical Weather Models etc.). Which of the following tropospheric products is the "true" reference?
6. The authors are not did a very rigorous job of reviewing the literature related to estimation ZTD (not water vapor) from multiple GNSS Observations. In the list of used literature directly on the topic of the article, there is only one article numbered 13 and from 2015. The rest of the articles are actually auxiliary. Has no one conducted research in this direction in the last five years.
Reviewer 3 Report
Comments and Suggestions for Authors
In this study, authors examined ZTD solutions for solo- and multi-GNSS solutions over 55 IGS stations. Furthermore, the results were compared with ERA5 and radiosonde data. According to their results, using four GNSS stations could improve the accuracy of the estimated ZTD. Because the GNSS meteorology community would find the study interesting, it is worth considering publishing it in the journal. In spite of this, I believe there are still some questions to be answered. As a result, I suggest a minor revision before publication.
General comments:
-My advice would be to read through the whole manuscript carefully and only use abbreviations that the readers are familiar with. Once they have been determined in the text, they should be used. Additionally, some abbreviations are not introduced in the manuscript, such as PPP in P2L79 or FES2004 in P2L88. An abbreviation need not be introduced if it is only mentioned once in the manuscript and not repeated like FES2004. The manuscript should be revised carefully in this regard.
-To make the entire manuscript more uniform, consider changing parentheses and notations to black, for example, Eq.7 and Eq.9.
Specific comments:
Abstract
The authors defined the abbreviation ZTD, but did not use it consistently thereafter. Thus, I suggest writing only ZTD once it is defined in the text.
Introduction
P1L35: Zenith dry delay (ZDD) could be changed to zenith hydrostatic delay (ZHD) as it is more familiar to GNSS users.P1L41: please provide two or three references for this statement.
P2L64: For consistency with other parts of the manuscript, please move [18] to in front of Zhang et al.
P2L74: The complete form of PANDA should be added, as well as some references regarding that.
Section 2:
P3L99: 55 Stations? Before reading subsection 3.1, the reader is not clear. For readers' benefit, please refer to Figure 1 or this subsection here.
P3L126: What is the reason for the noise intensity of 5-10 mm/h1/2. If possible, please provide an explanation and a reference.
Section 3.2:
Could authors provide a figure that illustrates radiosonde distribution, which would be helpful to readers in understanding this study better.
Section 4.1:
Figure 1: Please show the exact location of used two stations, namely DGAR and PTGG. What motivated the authors to investigate these two stations? Is there anything unique about these two stations in comparison to others?
Figure 2: What is the reason for this sudden jump in BDS-only for DOY about 260 (DGAR) and 200 (PTGG)? What is the most plausible explanation for that? Add a brief explanation in the associated part as well.
P7L210: There should be a space between the numbers and mm (e.g. 20 mm).
Figure 5: According to the RMSE, the differences between G-R and G-E appear to be almost comparable, except for a few stations. In spite of this, there are almost significant differences between G and C solutions, as well as G-C with two others. How could that be explained? Could it be a problem with BDS quality or ambiguity? In defense of this issue, please add a few sentences.
Section 4.2:
Figure 6: According to the results of DGAR stations, multi-GNSS solutions cannot guarantee high precision output in all epochs/days. As an example, there are huge fluctuations for GE, GCE, and GER on about DoY 270 in comparison with the one-system solution. What could be the cause of this? It needs to be explained in the manuscript as well.
Section 4.3:
According to Figure 9, the solutions for 2- and 3- systems are not significantly different. As a result, I suggest authors include an explanation of that in this and the conclusion sections.
Section 4.4:
It appears that there are no significant differences between 3- and 4-system solutions, according to the authors' reports. Moreover, the Bias and RMSE differences between 2- and 4-system are only 0.4 mm and 0.7 mm, respectively. It is therefore important for authors to provide some clarification here, as well as in the conclusion. Also, what are the plausible benefits of using 4-system solutions over, for example, 2-system solutions.
Table 3: The unit of RMSE and bias should be added here.
Conclusion:
Please update the conclusion based on the above comments.
Round 2
Reviewer 1 Report
Comments and Suggestions for Authors
This manuscript has been revised and I think it can be published in this journal.